# Repetitive DNA Dynamics, Phylogenetic Relationships and Divergence Times in Andean *Ctenomys* (Rodentia: Ctenomyidae)

**DOI:** 10.3390/biology14121776

**Published:** 2025-12-12

**Authors:** Rodrigo A. Vargas, Ronie E. Haro, Camilla Bruno Di-Nizo, Elkin Y. Suárez-Villota

**Affiliations:** 1Unidad de Producción Acuícola, Universidad de Los Lagos, Osorno 5290000, Chile; rvargasc@gmail.com; 2Department of Biochemistry and Molecular Biology, Dalhousie University, Halifax, NS B3H 4R2, Canada; ronie.haro@dal.ca; 3Leibniz Institute for the Analysis of Biodiversity Change, Museum Koenig, 53113 Bonn, Germany; camilladinizo@gmail.com; 4Instituto de Ciencias Naturales, Facultad de Medicina Veterinaria y Agronomía, Universidad de las Américas, Concepción 4030000, Chile

**Keywords:** chromosomal rearrangements, phylogenetics, speciation dynamics, comparative genomics, *Ctenomys maulinus*, Chile

## Abstract

This study aimed to determine the distribution and variation of repetitive DNA sequences in two closely related Andean *Ctenomys* species with different diploid numbers (2n = 26 and 2n = 28) using molecular cytogenetic approaches. Phylogenetic analyses were employed to infer the direction and timing of the detected chromosomal changes. The results suggest a reduction in repetitive DNA associated with chromosomal fission events that led to an increase in diploid number, with no evidence of recent major sequence turnover.

## 1. Introduction

Repetitive DNA constitutes nearly half of the nuclear genome in most eukaryotic species [1]. It is commonly divided into two categories: (i) highly repetitive sequences, or satellite DNA (satDNA), and (ii) moderately repetitive elements such as microsatellites, minisatellites, and transposable elements. satDNA, typically located in pericentromeric, subtelomeric, and interstitial heterochromatic regions, consists of tandemly repeated units that can extend into long arrays and is structurally dynamic, undergoing copy number changes via unequal crossing over, ectopic recombination, and replication errors, particularly during meiosis [2,3,4,5]. Such instability underlies chromosomal rearrangements, including translocations, fissions, and fusions, thereby reshaping genome architecture and promoting adaptation and speciation [6,7,8,9]. Empirical evidence supports a correlation between repetitive DNA and chromosomal variability in diverse organisms, including rodents [10,11]. Beyond their structural roles, satDNA sequences contribute to essential cellular functions, such as centromere formation, chromatin organization, spindle attachment, and gene regulation, through interactions with nuclear proteins and chromatin remodelling factors [12,13,14]. Owing to these functions and their fast evolutionary turnover, satDNA constitutes both a driver of genomic plasticity [15] and a valuable marker for tracing lineage divergence and genome evolution dynamics in natural populations [16,17].

Studies of the distribution and evolution of satDNA are especially informative in lineages with extensive karyotypic variability, such as the fossorial rodent genus *Ctenomys*. The genus represents one of the most prominent examples of radiation among South American mammals, occurring from Peru and Brazil to Tierra del Fuego, Argentina, and is composed of approximately 67 recognized species, allocated into nine distinct species groups [18,19]. Despite having a relatively conserved genome size [20,21], the genus displays an exceptionally high rate of chromosomal evolution, including recurrent amplifications and deletions of satellite DNA [22,23], and structural rearrangements such as Robertsonian translocations and pericentric inversions [24,25]. This karyotypic diversification is thought to be driven by isolation in small demes, genetic drift, and restricted dispersal associated with their subterranean lifestyle [26].

Within this framework, Andean species, hereafter referred to as the “*maulinus*” group, comprising *Ctenomys maulinus maulinus* (2n = 26; FNa = 48), *C. m. brunneus* (2n = 26; FNa = 48) and *Ctenomys* sp. (2n = 28; FNa = 50) [25,27]. The last two karyotypic forms occur parapatrically in the Lonquimay Valley, Chile, and their difference in diploid number led Gallardo [27] to suggest that the form with 2n = 28 represents an undescribed species. Unpublished studies [28,29] also point to a distinct taxonomic status for this cytotype; however, it is still considered a candidate species pending detailed molecular, morphological, and population-level analyses [18]. For this reason, we refer to the 2n = 28 form as *Ctenomys* sp. to maintain consistency with previous cytogenetic studies while acknowledging its unresolved taxonomic status. These karyotypic forms are characterized by homomorphic, uniarmed, medium-sized sex chromosomes, a rare condition in the genus, where biarmed and heteromorphic sex chromosomes are the rule [27,30]. In particular, *C. m. brunneus* and *Ctenomys* sp. share a high degree of molecular homology between their X and Y chromosomes [30]. This combination of divergence and similarity makes them a valuable model for exploring how repetitive DNA contributes to chromosomal differentiation and speciation, as shifts in satDNA arrays can reshape karyotypes and drive evolutionary change [1].

Therefore, in this study, we aimed to characterize the distribution and variation of highly repetitive DNA sequences in the Andean rodents *Ctenomys maulinus brunneus* and *Ctenomys* sp. using self-genomic in situ hybridization (Self-GISH) [31], whole-genomic comparative hybridization (W-CGH) [32,33], and fluorescent in situ hybridization (FISH) using a telomeric probe. These cytogenetic approaches allow us to identify the chromosomal regions involved in the gain or loss of repetitive DNA and to infer their potential role in karyotypic diversification. To contextualize these cytogenetic patterns within an evolutionary framework, we performed phylogenetic analyses and divergence time estimations based on partial cytochrome b (Cytb) sequences, providing temporal and directional evidence for the chromosomal changes detected. This integrative approach enables us to assess how repetitive DNA is distributed during the early stages of speciation and to evaluate its potential involvement in chromosomal differentiation between these closely related lineages.

## 2. Materials and Methods

### 2.1. Chromosomal Preparations and Genomic DNA Extraction

Mitotic plates from two males and two females of each *C. m. brunneus* (38°25′ S, 71°32′ W) and *Ctenomys* sp. (38°30′ S, 71°20′ W), both parapatrically distributed in Lonquimay, Valley, Chile, were analyzed. Metaphases were obtained from primary fibroblast cultures derived from lung tissue, following the protocol of Verma and Babu [34]. Genomic DNA was extracted from liver tissue using a phenol-chloroform method, as described by Sharma et al. [35]. Genomic DNA from two additional tissue samples from *C. m. maulinus* from Termas de Chillán (36°54′ S, 71°31′ W) were extracted for phylogenetic analysis. All samples were collected by M.H. Gallardo between 1972 and 2009 and are deposited in the Mammal Collection of the Universidad Austral de Chile, Valdivia, Chile, under the following field numbers: MHG1824 and MHG1826 for *C. m. brunneus*, MHG1823 and MHG567 for *Ctenomys* sp., and MHG986 and MHG981 for *C. m. maulinus*.

### 2.2. Cytogenetic Hybridization Techniques

To investigate the role of highly repetitive DNA in chromosomal evolution and speciation, self-genomic in situ hybridization (Self-GISH) [31] and whole-genomic comparative hybridization (W-CGH) [32,33] have emerged as powerful cytogenetic tools. The Self-GISH technique enables the detection of the distribution of highly repetitive DNA by hybridizing genomic DNA probes from a given species onto its own chromosomes. Similarly, W-CGH allows the identification of species-specific differences in the composition and distribution of repetitive elements by hybridizing labelled total genomic DNA from two species onto metaphase chromosomes. When coupled with differential fluorescence labelling, W-CGH facilitates the detection of divergent or conserved repetitive DNA families and highlights genomic regions that may have undergone expansion, contraction, or reorganization [33,36]. Given the central role of repetitive DNA in centromere function, chromatin structure, and genome plasticity, both Self-GISH and W-CGH provide effective approaches for linking repetitive DNA variation with chromosomal rearrangements and evolutionary processes [37,38,39,40].

To perform Self-GISH and W-CGH experiments, a red-labelled DNA probe from *Ctenomys* sp. and a green-labelled probe from *Ctenomys maulinus brunneus* were prepared using 12-UTP fluorescein and tetramethyl-rhodamine-5-dUTP, respectively (Roche Applied Science). For the Self-GISH assays, each probe was hybridized onto metaphase plates of its corresponding species (i.e., the *C. m. brunneus* probe on *C. m. brunneus* chromosomes). For the W-CGH experiments, both probes were mixed in equimolar concentrations and co-hybridized onto metaphase plates of *C. m. brunneus* in one assay, and onto those of *Ctenomys* sp. in a separate assay. All genomic in situ procedures were conducted according to the protocol described by Suárez-Villota et al. [39]. Telomeric sequences were detected using fluorescence in situ hybridization (FISH) on metaphase chromosome spreads, following the protocol described initially by Moyzis et al. [41]. PCR-generated probes containing the (TTAGGG)ₙ repeat motif were labelled with fluorescein-12-dUTP (Roche Applied Science, Penzberg, Germany), as outlined by Ijdo et al. [42]. For all in situ hybridization methods, mitotic plates were visualized under an Axiolab epifluorescence microscope (RE, 450905, Carl Zeiss, Oberkochen, Germany) equipped with an Axiocam camera. Fluorescence signals were captured using blue, red, and/or green filters (emission at 461, 570, and 517 nm, respectively), and image processing, including overlay and contrast enhancement, was performed using Adobe Photoshop v21.0.3.

### 2.3. Phylogenetic Analysis and Divergence Time Estimation

A previously published Cytb dataset for the genus *Ctenomys* reported by De Santi et al. [43] was used in combination with newly obtained sequences from *C. m. brunneus*, *C. m. maulinus*, and *Ctenomys* sp. to reconstruct phylogenetic relationships. Fragments of the Cytb gene were amplified using primers MVZ05 and MVZ16, following the PCR conditions described in Smith and Patton [44]. Macrogen Inc. conducted nucleotide sequencing, and all newly generated sequences were deposited in both GenBank and Zenodo (https://doi.org/10.5281/zenodo.17595543), under accession numbers PX551225–PX551230. Sequence quality was verified, and alignments were performed using Geneious v9.1.8 (https://www.geneious.com), employing the global pairwise alignment algorithm under the G-INS-i iterative refinement method. The best-fit codon nucleotide substitution models were identified using IQ-TREE v3.1 [45], and these were implemented in both maximum likelihood (ML) and Bayesian inference (BI) frameworks. The matrix was constructed using 76 terminal taxa and 1140 aligned sites. Outgroups were selected following De Santi et al. [43].

For ML analyses, three independent runs were performed using IQ-TREE v3.1 [45] to assess topological convergence. Nodal support was estimated through non-parametric bootstrapping with 1000 pseudoreplicates and SH-aLRT. For BI, we used MrBayes v3.2.6 [46], conducting two independent MCMC for 50,000,000 generations initiated from random trees, sampling every 1000 generations. Convergence and effective sample sizes were evaluated with Tracer v1.7.2 [47], and a burn-in was applied by discarding all samples collected prior to stationarity. The remaining trees were summarised to generate a majority-rule consensus tree, and branch support was estimated using Bayesian posterior probabilities.

For the divergence time analysis, we employed the same Cytb sequence dataset used in phylogenetic analyses. This dataset was analyzed using BEAST v2 [48], incorporating fossil calibrations based on *C. uquiensis* [49], *C. dasseni* [50], and *C. subassentiens*, together with their phylogenetic placements as reported by De Santi et al. [43]. Specifically, *C. uquiensis* was used to calibrate the stem of the *Ctenomys* lineage (lognormal prior: offset = 3.54 Ma, mean = 0.5, standard deviation = 0.7), *C. dasseni* was applied to the crown group of *Ctenomys* (offset = 0.75 Ma, mean = 0.4, standard deviation = 0.5), and *C. subassentiens* to the crown node of the clade composed of *C. frater*, *C. lewisi*, and *C. lessai* (offset = 0.75 Ma, mean = 0.3, standard deviation = 0.6). A lognormal relaxed molecular clock, a birth–death tree prior, and the HKY substitution model were employed, following the methodology described by De Santi et al. [43]. The analysis was performed using MCMC (Markov chain Monte Carlo) simulations for 100,000,000 generations, with parameters sampled every 10,000 generations. Convergence and effective sample sizes (ESS > 200) for all relevant parameters were assessed using Tracer v1.7.1 [47]. Following this, a 10% burn-in was applied before generating the maximum clade credibility (MCC) tree using TreeAnnotator.

## 3. Results

### 3.1. Cytogenetic Characterization of Repetitive Sequences

Self-GISH detected large pericentromeric regions containing highly repetitive sequences on eight chromosome pairs in *C. m. brunneus* (Figure 1a; pairs 1–8, green signals) and on four chromosome pairs in *Ctenomys* sp. (Figure 1b; pairs 4–7, red signals). In both species, the X chromosome exhibited a distinct pericentromeric region rich in repetitive sequences, enabling it to be differentiated from the Y chromosome. The Y chromosome, by contrast, exhibited only a small interstitial block of repetitive chromatin (Figure 1).

High molecular homology of repetitive sequences between *C. m. brunneus* and *Ctenomys* sp. were detected by W-CGH assays on their respective mitotic plates (Figure 2a,b; DAPI-stained chromosomes shown in the left panels). This homology was evidenced by the colocalization of fluorescent signals from the genomic DNA probes of *Ctenomys* sp. (red) and *C. m. brunneus* (green), displayed in the centre panels of Figure 2. The overlap of these signals produced a yellow–orange pattern, predominantly in the pericentromeric regions (Figure 2, right panels), indicating a high level of sequence similarity between the two species.

Telomeric, but no centromeric or interstitial signals were detected on all chromosomes of *C. m. brunneus* (Figure 3a) and *Ctenomys* sp. (Figure 3b).

### 3.2. Phylogeny and Divergence Times in Ctenomys

The codon model selected for the ML phylogenetic analyses was MGK + F3X4 + I + R4. The MGK model refers to the nonsynonymous/synonymous rate ratio (dN/dS) with an additional transition/transversion (ts/tv) ratio. F3X4 accounts for unequal nucleotide frequencies across the three codon positions. +I models a proportion of invariable sites, whereas +R4 represents rate heterogeneity among sites using four FreeRate categories. The best-scoring ML tree yielded a log-likelihood value of −10,030.7325. The Bayesian analysis recovered a consensus topology that closely resembled the ML tree, in which *C. m. maulinus*, *C. m. brunneus*, and *Ctenomys* sp. formed a well-supported monophyletic group hereafter referred to as the *maulinus* group (Figure 4: SH-aLRT = 100, bootstrap = 99; Figure 5: Bayesian posterior probability [BPP] = 1.0). Within this clade, *C. m. maulinus* diverged early, while *C. m. brunneus* and *Ctenomys* sp. were resolved as sister taxa (Figure 4: SH-aLRT = 70.3, bootstrap = 99; Figure 5: BPP = 0.93). Our analyses also recovered the nine previously recognized informal species groups: *frater*, *magellanicus*, *mendocinus*, *opimus*, *sociabilis*, *talarum*, *torquatus*, *boliviensis*, and *tucumanus* (Figure 4 and Figure 5). In the ML analyses, all groups were strongly supported (Figure 4; SH-aLRT > 90, bootstrap > 90), with the exception of the *boliviensis* group, which was not recovered as monophyletic, and the *opimus* group, which exhibited low support (Figure 4; SH-aLRT = 34.9, bootstrap = 75). The Bayesian analyses recovered all nine species groups as monophyletic with strong support (Figure 5; BPP > 0.98), except for the *mendocinus* group, which showed slightly lower support (Figure 5; BPP = 0.86). Topological discordance between methods was observed only in the case of the *boliviensis* group.

Estimates of divergence times based on cytochrome b sequences placed the most recent common ancestor (MRCA) of the genus *Ctenomys* in the early Pliocene, at approximately 3.13 million years ago (Ma), with a 95% highest posterior density (HPD) interval of 2.29–4.10 Ma (Figure 5). *Ctenomys osvaldoreigi* represents the earliest divergence within the genus, dating to the mid to late Pliocene. The *frater* group was estimated to have diverged at 2.51 Ma (95% HPD: 1.43–2.64 Ma), followed by the split between *C. leucodon* at 1.91 Ma (95% HPD: 1.31–2.60 Ma) and *C. tuconax* at 1.41 Ma (95% HPD: 0.92–1.96 Ma), preceding the diversification of the *sociabilis* group at 0.20 Ma (95% HPD: 0.04–0.21 Ma). A major node at 1.59 Ma (95% HPD: 1.12–2.10 Ma) represents the MRCA of several other informal groups: *boliviensis*, *magellanicus*, *maulinus*, *opimus*, *tucumanus*, *torquatus*, *talarum*, and *mendocinus*. These lineages diversified predominantly during the Late Pleistocene, with crown divergences ranging from 0.56 to 1.31 million years ago (Ma) (Figure 5).

The *maulinus* group, comprising *C. m. maulinus*, *C. m. brunneus*, and *Ctenomys* sp., diverged from its sister group at approximately 1.10 Ma (95% HPD: 0.73–1.92 Ma). Diversification within the *maulinus* group began at 0.42 Ma (95% HPD: 0.20–0.66 Ma), with the earliest split corresponding to *C. m. maulinus*, followed by the divergence between *C. m. brunneus* and *Ctenomys* sp. at 0.32 Ma (95% HPD: 0.15–0.51 Ma) (Figure 5; red-highlighted group). Overall, the chronogram indicates that most species-level diversification in *Ctenomys* is geologically recent, consistent with rapid radiations in Andean and southern South American environments.

## 4. Discussion

### 4.1. Phylogenetic Context and Karyotypic Evolution in the maulinus Group

To contextualize chromosomal differentiation in the *maulinus* group, we reconstructed phylogenetic relationships and estimated divergence times using the Cytb marker, allowing us to infer the direction and timing of chromosomal changes between the 2n = 26 and 2n = 28 karyotypes. Phylogenetic analyses based on the Cytb marker (Figure 4 and Figure 5) recovered most of the previously recognized *Ctenomys* species groups, i.e., [18,19,43,51]. However, the deeper nodes in the phylogeny, as well as the relationships among several species, remained poorly resolved, receiving low support values (Figure 4). This basal polytomy has traditionally been interpreted as the result of an explosive radiation, e.g., [52,53]. Additionally, a single molecular marker may not fully represent the species history owing to incomplete lineage sorting or introgression. In our reconstructions, *Ctenomys maulinus maulinus* (2n = 26) was recovered as the sister lineage to a well-supported monophyletic clade comprising *Ctenomys* sp. (2n = 28) and *C. m. brunneus* (2n = 26), which together form the *maulinus* group (Figure 4 and Figure 5). Based on outgroup comparison, phylogenetic topology, and differences in diploid number (see expanded view of the *maulinus* group in Figure 4), the karyotype with 2n = 28 was likely derived through chromosome fissions from an ancestral 2n = 26. In the analyzed karyotypes, no interstitial telomeric sequences (ITSs) were detected (Figure 3). Since ITSs are typically remnants of fusion events or telomeric insertions associated with double-strand break repair [54,55], their absence does not provide direct evidence for fission but is consistent with a scenario in which chromosomal breakage and reorganization occurred without detectable interstitial telomeric remnants. Moreover, a pericentric inversion or centromere repositioning in *C. m. brunneus* is suggested by the reduction in FN. Given that sex chromosomes are the only acrocentrics shared by both species, the autosomes likely represent the targets of these rearrangements, supporting a model of fissions and inversions driving karyotypic differentiation in the *maulinus* group.

To place the inferred chromosomal changes within a temporal and evolutionary framework, we estimated divergence times to identify when these rearrangements and the associated loss of repetitive sequences likely occurred, and to explore potential links with past environmental processes that could have influenced population dynamics. Our divergence time estimates are consistent with previous reports [43,51], which places the MRCA of the genus *Ctenomys* between the early Pliocene and the late Pleistocene. Most species groups appear to have originated during the late Early Pleistocene to the Middle Pleistocene (Figure 5). Notably, the origin and diversification of the *maulinus* group date to the Late Pleistocene, around 0.42 Ma (Figure 5). Thus, the evolutionary trajectory of lineages within this group likely unfolded under the influence of Quaternary climatic oscillations [56] and recurrent environmental disturbances, including volcanic activity [57,58]. These factors are known to drive demographic contractions and genetic bottlenecks in the *Ctenomys* genus [59,60]. Accordingly, these *Ctenomys* populations may represent a dynamic system characterized by cycles of local extinction and recolonization, potentially facilitating founder effects and shifts in the genetic background [60,61]. Such processes could also be associated with the chromosomal fission events and concomitant loss of repetitive sequences proposed here, which, according to general models, may contribute to the establishment of reproductive barriers and the fixation of chromosomal rearrangements through drift or selection [62,63].

### 4.2. Repetitive DNA Dynamics in Parapatric Ctenomys

Our molecular cytogenetic (Figure 1) and phylogenetic analyses (Figure 4 and Figure 5) allow us to infer a substantial loss of highly repetitive sequences during the chromosomal rearrangement events associated with the transition from 2n = 26 to 2n = 28 in the *maulinus* group. Chromosome rearrangements, including fission-fusion events, are often accompanied by the loss of repetitive elements, particularly those located in centromeric and telomeric regions, due to the inherent fragility and instability of these genomic sites [64]. Similar associations between satDNA turnover and chromosomal diversification have been reported in other *Ctenomys* lineages [22], with satDNA amplification and deletion being particularly relevant in sex chromosomes [23]. Indeed, although the X and Y chromosomes of both *C. m. brunneus* and *Ctenomys* sp. are homomorphic and difficult to distinguish by size or morphology using classical cytogenetic techniques [25,27], our molecular cytogenetic approach, based on repetitive sequence detection, allowed their clear identification (Figure 1). Altogether, these observations suggest that satDNA evolution is highly dynamic, contributing differently across chromosomal regions and playing a particularly important role in the origin and molecular homology of sex chromosomes, as previously discussed by Suárez-Villota et al. [30].

When comparing the distribution pattern of constitutive heterochromatin reported by Gallardo [25] with our results, although with low resolution, an interesting divergence emerges. In *C. m. brunneus*, heterochromatic regions are present in 11 autosomal pairs and the X chromosome, whereas repetitive DNA occurs in 8 pairs and the X chromosome. Similarly, in *Ctenomys* sp., constitutive heterochromatin is observed in 9 pairs and the X chromosome, while satDNA blocks are detected in 4 pairs and the X chromosome (compare Figure 1 with Appendix A). This pattern indicates that not all satDNA in *Ctenomys* co-localizes with C-positive blocks, challenging the assumption of a direct correspondence between these elements. In this context, the reduction of repetitive sequences in the 2n = 28 karyotype appears to be associated with chromosomal fission events underlying the increase in diploid number, as supported by the phylogenetic topology, but not directly with heterochromatin loss. The lack of perfect correspondence between satDNA and C-positive blocks suggests that heterochromatin formation may also involve other repetitive elements occupying these regions, or that heterochromatic states could be epigenetically maintained following sequence loss or turnover. Similar patterns have been observed in several non-model species, highlighting that the relationship between satellite DNA and heterochromatin is not universal but context-dependent (e.g., [65]). Moreover, recent analyses of “satellitomes” have revealed that satellite DNA organization and evolution are far more dynamic and complex than previously thought (reviewed in [66]).

Despite quantitative differences in the chromosomal distribution of highly repetitive sequences between *Ctenomys* sp. and *C. maulinus brunneus*, W-CGH revealed a high degree of molecular homology between these elements (Figure 2). This finding suggests that sequence-level divergence has remained limited, possibly due to a relatively low mutation rate and the short evolutionary time since the divergence of these taxa (~0.32 Ma). In such cases, repetitive elements may vary rapidly in abundance or chromosomal position while remaining conserved at the sequence level. Similar patterns have been reported in other rodents [67,68], and it is likely that recent divergence or homogenizing mechanisms, such as gene conversion, contribute to the maintenance of sequence identity among repeat units [17,69,70]. Moreover, the observed sequence conservation may reflect functional constraints related to the regulatory roles of satellite DNA. Recent studies have shown that satellite repeats can influence gene expression, chromatin organization, and stress responses, particularly when dispersed within euchromatin or transcribed into regulatory RNAs, thus contributing to the stability and maintenance of specific satellite families despite their repetitive nature [71]. Altogether, these results imply that amplification and redistribution of repetitive sequences may precede their sequence differentiation, particularly in lineages undergoing recent chromosomal diversification.

## 5. Conclusions

Our results provide new insights into the interplay between chromosomal evolution and the dynamics of repetitive DNA in the *Ctenomys maulinus* group. Phylogenetic and divergence time analyses support a recent diversification scenario shaped by Quaternary environmental fluctuations. Both cytogenetic and phylogenetic evidence indicate a directional transition from a 2n = 26 to a 2n = 28 karyotype, likely involving fission events in addition to pericentric inversion or centromere repositioning that could also explain the FN difference. This transition is accompanied by a reduction in highly repetitive sequences, which reflects the relationship between the distribution of unstable genomic regions and karyotypic stability. Despite quantitative differences in repeat distribution, the high sequence homology observed between taxa suggests that mechanisms such as gene conversion, together with limited divergence time, may constrain sequence-level differentiation. Overall, this study highlights the importance of integrating molecular cytogenetics and phylogenetics to examine how genome architecture evolves in response to structural, demographic, and environmental factors in subterranean mammals.

## Figures and Tables

**Figure 1 biology-14-01776-f001:**
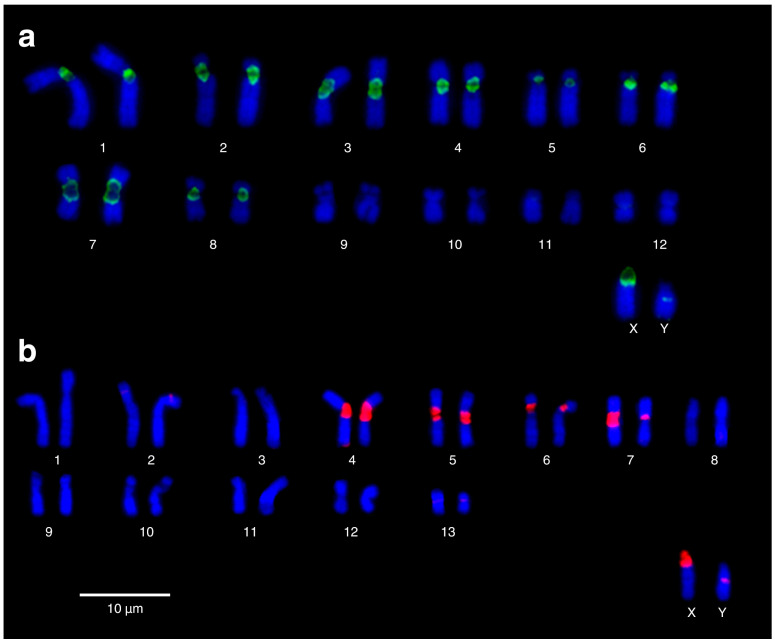
Self-GISH on mitotic metaphase spreads. (**a**) *Ctenomys maulinus brunneus* (2n = 26; FNa = 48); (**b**) *Ctenomys* sp. (2n = 28, FNa = 50). Large blocks of highly repetitive sequences are detected in the pericentromeric regions, labelled in green and red, respectively. Signals are present on the X chromosomes and several autosomes in both species. See main text for details.

**Figure 2 biology-14-01776-f002:**
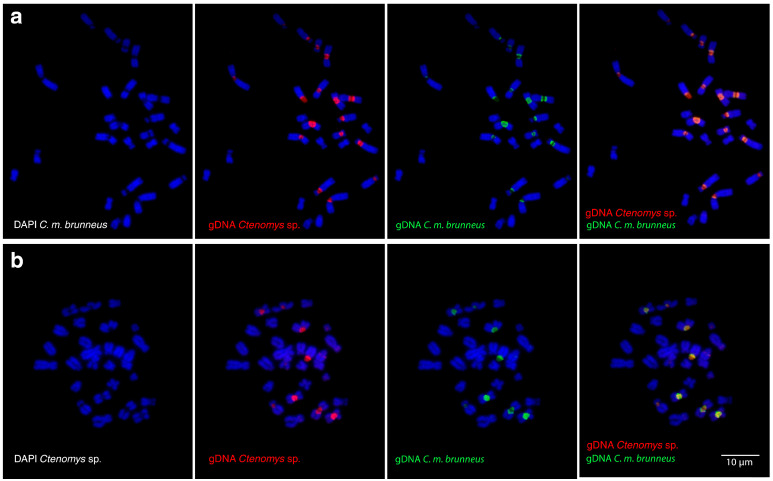
Whole-genomic comparative hybridization (W-CGH) between *Ctenomys maulinus brunneus* and *Ctenomys* sp. (**a**) Metaphase plate of *C. m. brunneus* co-hybridised with genomic DNA from *Ctenomys* sp. (red) and *C. m. brunneus* (green). (**b**) Metaphase plate of *Ctenomys* sp. hybridized with the same DNA mixture as in (**a**). In both cases, DAPI-stained chromosomes (blue channel) are shown in the left panels, followed by the individual fluorescence signals of *Ctenomys* sp. (red) and *C. m. brunneus* (green) in the centre panels. The merged images (right panels) display yellow–orange signals in the pericentromeric regions, indicating colocalization and high molecular homology of repetitive sequences between the two species.

**Figure 3 biology-14-01776-f003:**
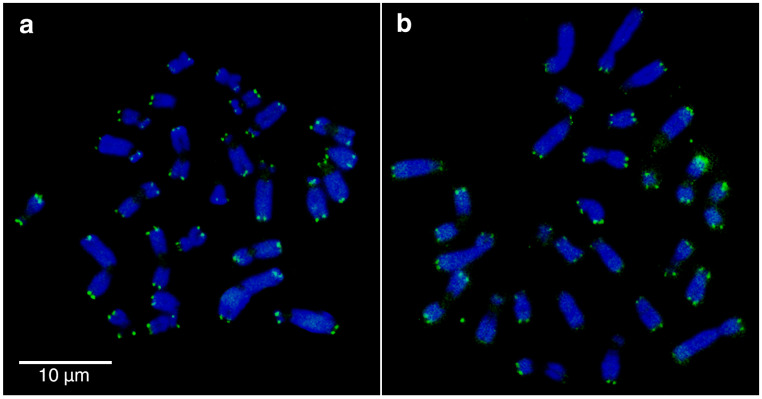
FISH using a telomeric probe on mitotic plates from male individuals. (**a**) *Ctenomys maulinus brunneus* and (**b**) *Ctenomys* sp. Telomeric signals (in green) are restricted to terminal regions, with no evidence of interstitial telomeric sequences (ITS) detected on any chromosome.

**Figure 4 biology-14-01776-f004:**
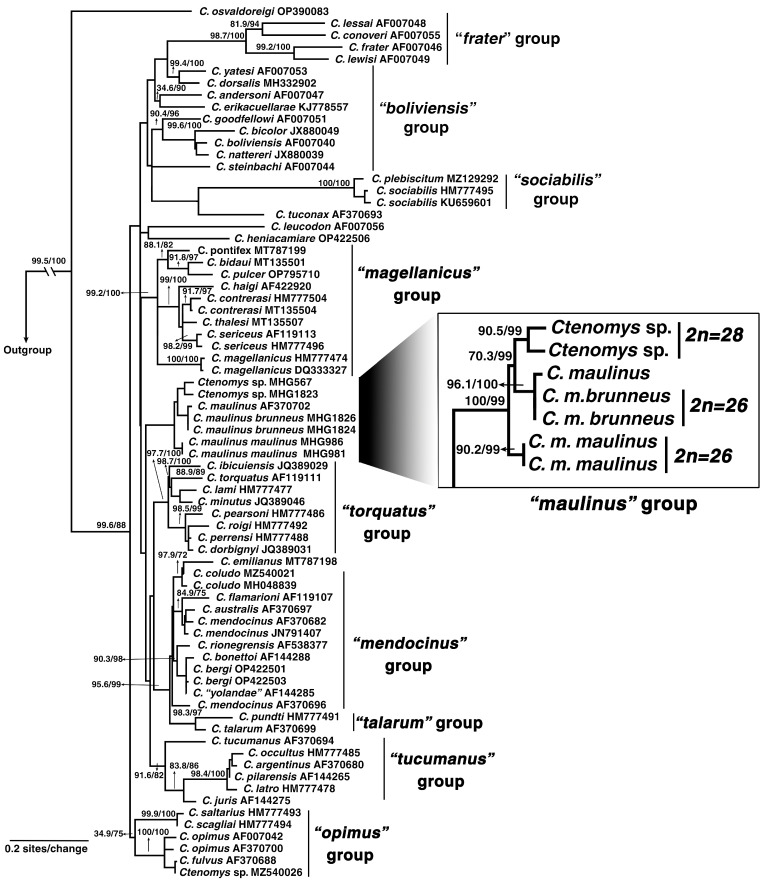
Phylogenetic relationships within *Ctenomys* based on cytochrome b sequences. Maximum likelihood tree including *C. maulinus* and *Ctenomys* sp. individuals. Nodal support values are shown above the branches as SH-aLRT/Bootstrap percentages. Informal species groups reported by Teta et al. [19] are indicated. The “*maulinus*” group is shown in an **expanded view**, with diploid numbers indicated. Note the monophyly of the group and the early divergence of *C. m. maulinus*.

**Figure 5 biology-14-01776-f005:**
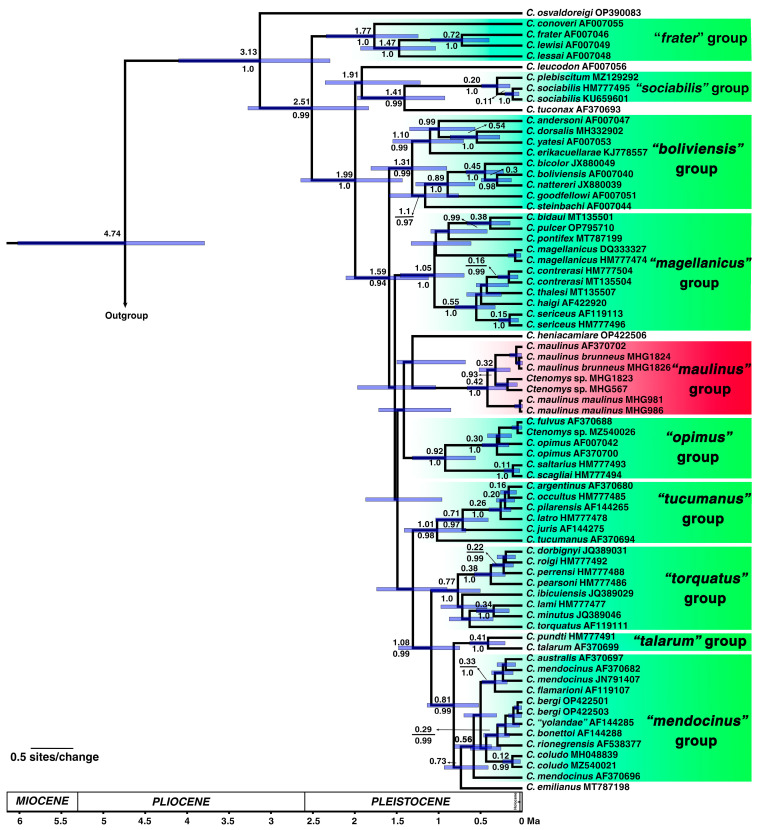
Divergence time estimates within *Ctenomys* based on cytochrome b sequences. Chronogram inferred using a relaxed molecular clock under a Bayesian framework. Numbers above branches indicate divergence times in millions of years (Ma), and values below branches represent Bayesian posterior probabilities. Blue bars at nodes denote 95% highest posterior density (HPD) intervals. GenBank accession numbers and informal species groups reported by Teta et al. [19] are indicated to the right of the species names. Note the late Pleistocene diversification of the “*maulinus*” group, highlighted in red.

## Data Availability

The original contributions presented in this study are included in the article. Further inquiries can be directed to the corresponding author.

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
