# Peer review of "Biology2025, 14(12), 1776;https://doi.org/10.3390/biology14121776"

_biology, 2025, doi:10.3390/biology14121776_

Round 1

Reviewer 1 Report

Comments and Suggestions for Authors

This paper focuses on the dynamics of repetitive DNA, phylogenetic relationships and differentiation times. The research design is reasonable and the data support is relatively sufficient. However, minor adjustments need to be made in terms of method details and result interpretation.but there are still several issues that need to be addressed.

1.MGK + F3X4 + I + R4 is merely a string. Could you please explain in the text what the model partition represented by MGK + F3X4 + I + R4 is ?

2.Can it be determined whether the high-repetitive sequences detected actually belong to satellite DNA, recombinant DNA or LINE?

3.In the ML analysis, "the group support of opimus is low (SH-aLRT = 34.9, bootstrap = 75)". The reasons for the low support should be explained in the discussion.

Author Response

This paper focuses on the dynamics of repetitive DNA, phylogenetic relationships and differentiation times. The research design is reasonable and the data support is relatively sufficient. However, minor adjustments need to be made in terms of method details and result interpretation.but there are still several issues that need to be addressed.

Comment 1: MGK + F3X4 + I + R4 is merely a string. Could you please explain in the text what the model partition represented by MGK + F3X4 + I + R4 is?

Response 1: Codon substitution rates and frequencies were calculated in IQ-TREE, which supports several codon models. MGK refers to nonsynonymous/synonymous (dn/ds) rate ratio (Muse and Gaut, 1994), with additional transition/transversion (ts/tv) rate ratio. F3X4 refers to unequal nucleotide frequencies and unequal nt frequencies over three codon positions. I designates the rate heterogeneity across sites and allows for a proportion of invariable sites while R4 means 4 FreeRate categories which use four categories (or bins) representing different substitution rates. The explanation was included in the manuscript text (lines 228-231).

Comment 2: Can it be determined whether the high-repetitive sequences detected actually belong to satellite DNA, recombinant DNA or LINE?

Response 2: We agree with the reviewer since GISH / CGH allows the detection of highly repetitive genomic regions but does not distinguish the specific type of repeats (e.g., satellite DNA or transposable elements). Further sequence-based analyses would be necessary for such classification.

Comment 3: In the ML analysis, "the group support of opimus is low (SH-aLRT = 34.9, bootstrap = 75)". The reasons for the low support should be explained in the discussion.

Response 3: The low support recovered for the opimus group may be attributed to the use of sequences from a single gene (mitochondrial Cytb) and to the incomplete taxonomic sampling, as not all representatives of the genus were included in the analysis (a total of 67 species is currently recognized for the genus). This limited resolution likely reflects the short divergence times associated with recent speciation events. Moreover, relying on a single molecular marker prevents ruling out errors arising from incomplete lineage sorting or introgression, which may lead to incongruences between gene trees and the species tree. As suggested, we added one sentence with this explanation in the discussion (lines 293-295).

Reviewer 2 Report

Comments and Suggestions for Authors

Manuscript ID:biology-3945796

Title:Repetitive DNA Dynamics, Phylogenetic Relationships and Divergence Times in Andean Ctenomys (Rodentia: Ctenomyidae)

Rodrigo A. Vargas, Ronie E. Haro, Camilla Bruno Di-Nizo, Elkin Y. Suárez-Villota

In this study, the authors use various phylogenetic and chromosomal analysis to study several species of Ctenomys. Why the study appears to be scientifically sounds, it is somewhat unclear why the analyses were performed, and overall, what the aim of this study was. It seems from the "Simple Summary"  that the study focuses on repetitive sequences in chromosome evolution, yet the authors have performed a variety of phylogenetic analyses, including estimation of divergence between the species. The authors should clearly explain to the reader what exactly the aim of the study was, what the aims of the various analyses were, and how conclusions were draw. Below are some specific points that need to be addressed (some minor, other major):

1. In the Abstract, Fundamental number is abbreviated as FN and NF. I believe it should be FN, but if both are acceptable, then  one should probably be used consistently. 

2. "Ctenomys sp." in the abstract as well as the document itself is very confusing because it is unusual to have a genus with no labeled species name and the authors do not say why this is the case. Is this an unknown specimen that the authors are focusing on to better understand who it is related to and how it is different than the other two known species?? 

It seems like the entire study is focusing on this Ctenomys sp.   but the fact that it is of an unknown species is not addressed by the authors at all. The authors should explain this better, introduce this specimen(s) as one that does not have a species name or is still undescribed, or whatever the case is. 

3. Section 4.1. Why did the authors build a Cytochrome B phylogenetic tree? They even cited 4 references where this was already done with ref #19 even used Cytochrome B. Did they do this to figure out who the “Ctenomys sp.” was related to? if so, they should state this explicitly, if not, they should say why the went through the trouble of doing this since the phylogenetic relationships of the major groups of this genus have been worked on by others already. 

4.) Section 4.1. - lines 275-277 - While it is clear how chromosomal fusion can created ITSs, it is less clear how the "absence of ITSs strengthens the likelihood of chromosomal fission having taken place in Ctenomys". Could the authors expand on this more? 

5.) Section 4.1. - lines 282 - 284 - as noted above, it is unclear how absence of ITSs strengthens the likelihood of chromosomal fission. However, in this is subsequent section, the authors further theorize how ITSs may be lost due to elimination of sequence due to chromosome breaks. While this is all true, in theory, it is all conjecture... no data is provided to prove that this has happened. 

6.) Section 4.1. - lines 282 what do the authors mean by "phylogenetically inferred direction of change." Please rephrase. 

7.)  If the authors are interested in finding homology in chromosomes between Ctenomys maulinus brunneus and Ctenomys sp., why not do chromosomal banding? C or G banding? this might help to understand where the changes happened. 

8.) In Section 4.2 - lines 310 - 314 - the authors discuss comparison to previous work done by Gallardo et al (Reference 25), where they discuss overlap between repetitive DNA and heterochromatic region overlap. Perhaps the authors could make an illustration of this and add it to Figure 3? This seems very relevant and putting some effort into communicating this to their reader would go a long way. 

9.) Lastly, it is unclear once again why the authors performed divergence date analysis. Has this not been done before? The authors cite several previous reports, so it's not clear why they would repeat this. Again, some explanation of this would be very helpful in explaining their reasoning to the reader. 

Author Response

General comment: In this study, the authors use various phylogenetic and chromosomal analysis to study several species of Ctenomys. Why the study appears to be scientifically sounds, it is somewhat unclear why the analyses were performed, and overall, what the aim of this study was. It seems from the "Simple Summary"  that the study focuses on repetitive sequences in chromosome evolution, yet the authors have performed a variety of phylogenetic analyses, including estimation of divergence between the species. The authors should clearly explain to the reader what exactly the aim of the study was, what the aims of the various analyses were, and how conclusions were draw. Below are some specific points that need to be addressed (some minor, other major):

Response: We have revised the Simple Summary (lines 14-19), Abstract (lines 21-36), and Introduction (lines 90-102) to clearly state the main aim of the study and the purpose of each analysis. Specifically, we clarified that the objective was to determine the distribution and variation of repetitive DNA sequences using molecular cytogenetic approaches, and to use phylogenetic analyses to infer the timing and direction of the cytogenetic changes observed.

Additionally, we have reorganised the methods and results sections to present the molecular cytogenetic analyses first, where the distribution of repetitive sequences is described, followed by the phylogenetic analyses. This new order provides a clearer logical flow and aligns with the sequence in which the study objectives were originally formulated.

Comment 1: In the Abstract, Fundamental number is abbreviated as FN and NF. I believe it should be FN, but if both are acceptable, then one should probably be used consistently.

Response 1: We thank the reviewer for this observation. The abbreviation has been standardized throughout the manuscript, and NF has been replaced by FN in all instances.

Comment 2: "Ctenomys sp." in the abstract as well as the document itself is very confusing because it is unusual to have a genus with no labeled species name and the authors do not say why this is the case. Is this an unknown specimen that the authors are focusing on to better understand who it is related to and how it is different than the other two known species??

It seems like the entire study is focusing on this Ctenomys sp.  but the fact that it is of an unknown species is not addressed by the authors at all. The authors should explain this better, introduce this specimen(s) as one that does not have a species name or is still undescribed, or whatever the case is.

Response 2: We appreciate the reviewer’s observation regarding the use of Ctenomys sp. in the manuscript. In response, we have revised the introduction (lines 73–82) to clarify that this taxon represents an undescribed form with 2n = 28, previously reported by Gallardo et al 1979, which occurs parapatrically with C. m. brunneus in the Lonquimay Valley, Chile. Unpublished studies (Aguilar, 1988; Ramírez, 2017) suggest that this cytotype corresponds to distinct species, but they are still considered candidate species pending integrative taxonomic analyses. Therefore, we refer to the 2n = 28 form as Ctenomys sp. to maintain consistency with previous cytogenetic studies while acknowledging its unresolved taxonomic status. This information has been added to justify the use of Ctenomys sp. throughout the text.

Comment 3: Section 4.1. Why did the authors build a Cytochrome B phylogenetic tree? They even cited 4 references where this was already done with ref #19 even used Cytochrome B. Did they do this to figure out who the “Ctenomys sp.” was related to? if so, they should state this explicitly, if not, they should say why the went through the trouble of doing this since the phylogenetic relationships of the major groups of this genus have been worked on by others already.

Response 3: We thank the reviewer for this observation. As now clarified in the Introduction (lines 96–99) and in Section 4.1 (lines 285-288, 310-313), the phylogenetic analyses were not intended to reconstruct the entire genus Ctenomys (as done in previous studies), but rather to provide a temporal and evolutionary framework for interpreting the chromosomal changes detected between C. m. brunneus (2n = 26) and Ctenomys sp. (2n = 28).
In addition, the revised Figure 4 now presents an expanded view of the “maulinus group” clade, with diploid numbers (2n) indicated, highlighting its monophyly and the early divergence of C. m. maulinus. This integration of phylogenetic and cytogenetic data provides a more robust framework for interpreting the direction and timing of karyotypic evolution within the group.

Comments 4 and 5: Section 4.1. - lines 275-277 - While it is clear how chromosomal fusion can created ITSs, it is less clear how the "absence of ITSs strengthens the likelihood of chromosomal fission having taken place in Ctenomys". Could the authors expand on this more?

Section 4.1. - lines 282 - 284 - as noted above, it is unclear how absence of ITSs strengthens the likelihood of chromosomal fission. However, in this is subsequent section, the authors further theorize how ITSs may be lost due to elimination of sequence due to chromosome breaks. While this is all true, in theory, it is all conjecture... no data is provided to prove that this has happened.

Response 4 and 5: We thank the reviewer for this valuable observation. We agree that the absence of interstitial telomeric sequences (ITSs) cannot be considered direct evidence of chromosomal fission, as these sequences may also be lost or remain undetectable after rearrangements. We have therefore modified this section (lines 300–305) to present the absence of ITSs as an observational result rather than as a causal inference, emphasizing that the direction of chromosomal change (fission hypothesis) is primarily supported by the phylogenetic topology and the differences in diploid number among the analyzed species.

Comment 6: Section 4.1. - lines 282 what do the authors mean by "phylogenetically inferred direction of change." Please rephrase.

Response 6: It was rewrite (line 300)

Comment 7:  If the authors are interested in finding homology in chromosomes between Ctenomys maulinus brunneus and Ctenomys sp., why not do chromosomal banding? C or G banding? this might help to understand where the changes happened.

Response 7: We thank the reviewer for this valuable suggestion. However, the main goal of our study was not to establish full chromosomal homology but rather to analyse the distribution and variation of repetitive DNA sequences using molecular cytogenetic techniques (Self-GISH, W-CGH, and FISH). Conventional G-banding was not showed because, despite obtaining high-quality metaphase spreads suitable for molecular analyses, G-band patterns in Ctenomys were typically faint and difficult to reproduce, limiting their usefulness for comparative purposes. We consider that more advanced approaches, such as chromosome painting or cross-species FISH (Zoo-FISH), would be more informative for future studies aimed at establishing detailed chromosomal homology between C. m. brunneus and Ctenomys sp.

Comment 8: In Section 4.2 - lines 310 - 314 - the authors discuss comparison to previous work done by Gallardo et al (Reference 25), where they discuss overlap between repetitive DNA and heterochromatic region overlap. Perhaps the authors could make an illustration of this and add it to Figure 3? This seems very relevant and putting some effort into communicating this to their reader would go a long way.

Response 8: We appreciate the reviewer’s thoughtful suggestion. We agree that an illustration could help readers visualize the relationship between heterochromatic and repetitive regions. However, we cannot be certain that the chromosomes analyzed by C-banding in Gallardo (1991) correspond precisely to those examined in our molecular cytogenetic experiments. Ideally, both analyses should have been performed on the same metaphase plates to establish a direct correspondence, but unfortunately, the material is no longer available to repeat the experiments. Including the idiogram alongside Figure 1 could therefore be misleading, as several chromosomal pairs in Ctenomys have similar sizes and morphologies, preventing unambiguous identification.
Nonetheless, to aid interpretation and improve clarity, we prepared an idiogram following the order and heterochromatic pattern presented by Gallardo (1991), including it as Figure S1 (Line 397). This addition allows readers to compare the distribution of heterochromatic regions and repetitive sequences, while acknowledging the limitations in chromosome correspondence (Lines 346-352).

Comment 9: Lastly, it is unclear once again why the authors performed divergence date analysis. Has this not been done before? The authors cite several previous reports, so it's not clear why they would repeat this. Again, some explanation of this would be very helpful in explaining their reasoning to the reader.

Response 9: We appreciate the reviewer’s observation. In the revised manuscript, we now clarify in section 4.1 (lines 310–313) that divergence time estimation was conducted to place the chromosomal changes and the loss of repetitive sequences in a temporal and evolutionary context. This analysis allowed us to associate the inferred cytogenetic events with Late Pleistocene environmental dynamics, thereby integrating phylogenetic and cytogenetic evidence to better understand the evolutionary processes shaping the maulinus group.

Reviewer 3 Report

Comments and Suggestions for Authors

The paper describes a study of repetitive sequences in two parapatric Andean taxa of the genus Ctenomys from Chile using a set of in situ hybridization techniques. The data were combined with a phylogenetic analysis and dating based on cytochrome b sequences. In this way the authors were able to show a loss of highly repeated sequences during the transition from 2n = 26 to 28 while there have been no recent changes in the sequences.

I found the conclusion “that not all satDNA in Ctenomys co-localizes with C-positive blocks, challenging the assumption of a direct correspondence between these elements” interesting. But, at the same time, I miss some explanation how it can happen that repetitive sequences have been lost without a loss of heterochromatin, when these repetitive sequences are heterochromatic. Therefore, the authors should discuss this question in more detail. Moreover, the authors should be more explicit in their conclusions about the evolution within the group under study. In particular, the text implies that the karyotype of Ctenomys sp. (2n = 28; NF = 50) has originated through a fission from that of Ctenomys maulinus brunneus (2n = 26; FN = 48), which is improbable given the different number of chromosome arms. A more plausible scenario is the origin from C. m. maulinus (2n = 26; FN = 50), which is consistent with the basal position of the latter taxon within the group. So, the authors should be more explicit in this respect. In addition, if this scenario is correct, the authors should address the question about the evolution of C. m. brunneus maintaining the ancestral 2n but decreasing FN.

Below, I show several remarks and questions:

Line 78: The assertion that the sex chromosomes of the species under study are monomorphic, doesn´t seem to agree with Figure 3 a and b.

L121: How many Markov chains were run?

L181: “monophyletic clade” – clade = monophylum; therefore, use either “monophyletic group” or “clade” only.

L185: In my opinion, bootstrap support = 99 and posterior probability = 0.93 are rather high values, so I don´t think the support is “moderate”.

L189–190: Italicize “boliviensis” and “opimus”.

L177–194, Figures 1–2: There are two taxa, C. leucodon and C. heniacamiare, which are problematic: they don´t fall into any clade and their placement differs in ML and Bayesian phylogenies.

L274–275: “Based on outgroup comparison, the karyotype with 2n = 28 was likely a result of chromosome fissions.” – see my comment above. It should be expressed more explicitly which taxon has originated from which (given the main focus of the study are C. m. brunneus and Ctenomys sp., this assertion implies a wrong conclusion). An alternative explanation would be a fission inside an arm of brunneus, e.g. between previous telomere and centromere, but I´m not aware of any such event somewhere else.

L278: I would add a centromeric-telomeric fusion (see the origin of human chromosome 2).

Author Response

Comment 1: The paper describes a study of repetitive sequences in two parapatric Andean taxa of the genus Ctenomys from Chile using a set of in situ hybridization techniques. The data were combined with a phylogenetic analysis and dating based on cytochrome b sequences. In this way the authors were able to show a loss of highly repeated sequences during the transition from 2n = 26 to 28 while there have been no recent changes in the sequences.

I found the conclusion “that not all satDNA in Ctenomys co-localizes with C-positive blocks, challenging the assumption of a direct correspondence between these elements” interesting. But, at the same time, I miss some explanation how it can happen that repetitive sequences have been lost without a loss of heterochromatin, when these repetitive sequences are heterochromatic. Therefore, the authors should discuss this question in more detail.

Response 1: We appreciate the reviewer’s insightful comment. We agree that the apparent discrepancy between the reduction of certain repetitive sequences and the persistence of heterochromatic regions warrants clarification. In our study, the observation that not all satDNAs co-localize with C-positive blocks suggests that heterochromatin composition and maintenance are more complex than previously assumed. It is plausible that (i) other repetitive elements, such as transposable elements or microsatellites, contribute to the C-positive heterochromatic regions, or (ii) epigenetic mechanisms, including DNA methylation and histone modifications, preserve the heterochromatic state even after partial sequence loss or turnover of specific satDNA families. We have expanded the discussion section (lines 357–364) to include these explanations and incorporated two additional references ( Rico-Porras et al., 2024; Šatović-Vukšić et al., 2023) that provide recent evidence of similar patterns in non-model species and highlight the growing complexity of satellite DNA organization in eukaryotic genomes.

Rico-Porras, J.M.; Mora, P.; Palomeque, T.; Montiel, E.E.; Cabral-de-Mello, D.C.; Lorite, P. Heterochromatin is not the only place for satDNAs: the high diversity of satDNAs in the euchromatin of the Beetle Chrysolina americana (Coleoptera, Chrysomelidae). Genes 2024, 15, https://doi.org/10.3390/genes15040395.

Šatović-Vukšić, E.; Plohl, M. Satellite DNAs—from localized to highly dispersed genome components. Genes 2023, 14, https://doi.org/10.3390/genes14030742.

Comment 2: Moreover, the authors should be more explicit in their conclusions about the evolution within the group under study. In particular, the text implies that the karyotype of Ctenomys sp. (2n = 28; NF = 50) has originated through a fission from that of Ctenomys maulinus brunneus (2n = 26; FN = 48), which is improbable given the different number of chromosome arms. A more plausible scenario is the origin from C. m. maulinus (2n = 26; FN = 50), which is consistent with the basal position of the latter taxon within the group. So, the authors should be more explicit in this respect. In addition, if this scenario is correct, the authors should address the question about the evolution of C. m. brunneus maintaining the ancestral 2n but decreasing FN.

Response 2: We appreciate the reviewer’s insightful comment and the opportunity to clarify this important point. The confusion regarding the fundamental number (FN) indeed stems from the inclusion of the sex chromosome pair when the karyotype of C. m. maulinus was originally described, which has been inconsistently reported in subsequent studies. We thank the reviewer for highlighting this issue, as it has been perpetuated in serial publications since the first karyotypic description of C. m. maulinus.

To avoid ambiguity, we have now corrected the notation throughout the manuscript, replacing FN with FNa (autosomal fundamental number). Accordingly, C. m. maulinus and C. m. brunneus share the same autosomal configuration (2n = 26, FNa = 48), clarifying that their difference from Ctenomys sp. (2n = 28, FNa = 50) involves not only chromosomal fission but also a probable pericentric inversion or centromere repositioning event.

These clarifications are now explicitly incorporated into the text (lines 305, 306) and the Conclusions section (lines 387-389), to make the inferred chromosomal evolutionary scenario of the “maulinus” group clearer and more consistent

Comment 3: Line 78: The assertion that the sex chromosomes of the species under study are monomorphic, doesn´t seem to agree with Figure 3 a and b.

Response 3: We described the sex chromosomes as homomorphic because, in all three species analyzed, they are uniarmed and of similar medium size, making them difficult to distinguish based solely on morphology in conventional cytogenetic preparations. However, as now clarified in the Discussion, our molecular cytogenetic analyses based on repetitive sequence detection allowed us to clearly differentiate the X and Y chromosomes in C. m. brunneus and Ctenomys sp. (see revised text, lines 338–342 and Figure 1).

Comment 4: L121: How many Markov chains were run?

Response 4: Two independent MCMC  for 50,000,000 generations were run. We added this information in line 168.

Comment 5: L181: “monophyletic clade” – clade = monophylum; therefore, use either “monophyletic group” or “clade” only.

Response 5: We agree and corrected the term in the text (line 234).

Comment 6: L185: In my opinion, bootstrap support = 99 and posterior probability = 0.93 are rather high values, so I don´t think the support is “moderate”.

Response 6: We agree and corrected in the manuscript text (line 237).

Comment 7: L189–190: Italicize “boliviensis” and “opimus”.

Response 7: The group names were italicized. Thank you (lines 242, 243).

Comment 8: L177–194, Figures 1–2: There are two taxa, C. leucodon and C. heniacamiare, which are problematic: they don´t fall into any clade and their placement differs in ML and Bayesian phylogenies.

Response 8: Indeed, both taxa have low phylogenetic resolution but we believe that this do not affect the discussion about chromosomal evolution within “malinus” group.

Comment 9: L274–275: “Based on outgroup comparison, the karyotype with 2n = 28 was likely a result of chromosome fissions.” – see my comment above. It should be expressed more explicitly which taxon has originated from which (given the main focus of the study are C. m. brunneus and Ctenomys sp., this assertion implies a wrong conclusion). An alternative explanation would be a fission inside an arm of brunneus, e.g. between previous telomere and centromere, but I´m not aware of any such event somewhere else.

Response 9: We agree that the process of chromosomal evolution is more complex than we pointed out in the first version of this manuscript. We believe that other rearrangements like pericentric inversion / centromere repositioning probably explain the FN differences and we included this information in the text (lines 305-309, 387-389).

Comment 10: L278: I would add a centromeric-telomeric fusion (see the origin of human chromosome 2).

Response 10: We respectfully disagree. The origin of human chromosome 2 resulted from a telomere–telomere fusion, not a centromeric–telomeric fusion, as demonstrated by Ijdo et al. (1991, PNAS, 88:9051-5).

Round 2

Reviewer 2 Report

Comments and Suggestions for Authors

The authors have addressed all of my previous concerns in detail. I have no further comments.